# Carbonic Anhydrase IX—Mouse versus Human

**DOI:** 10.3390/ijms21010246

**Published:** 2019-12-30

**Authors:** Martina Takacova, Monika Barathova, Miriam Zatovicova, Tereza Golias, Ivana Kajanova, Lenka Jelenska, Olga Sedlakova, Eliska Svastova, Juraj Kopacek, Silvia Pastorekova

**Affiliations:** Biomedical Research Center, Institute of Virology, Department of Tumor Biology, Slovak Academy of Sciences, Dubravska cesta 9, 84505 Bratislava, Slovakia; martina.takacova@savba.sk (M.T.); monika.barathova@savba.sk (M.B.); miriam.zatovicova@savba.sk (M.Z.); tereza.golias@gmail.com (T.G.); iva.kajanova@gmail.com (I.K.); virulesu@savba.sk (L.J.); juraj.kopacek@savba.sk (J.K.)

**Keywords:** carbonic anhydrase IX, transcriptional regulation, pH regulation, migration, exosomes, hypoxia

## Abstract

In contrast to human carbonic anhydrase IX (hCA IX) that has been extensively studied with respect to its molecular and functional properties as well as regulation and expression, the mouse ortholog has been investigated primarily in relation to tissue distribution and characterization of CA IX-deficient mice. Thus, no data describing transcriptional regulation and functional properties of the mouse CA IX (mCA IX) have been published so far, despite its evident potential as a biomarker/target in pre-clinical animal models of tumor hypoxia. Here, we investigated for the first time, the transcriptional regulation of the *Car9* gene with a detailed description of its promoter. Moreover, we performed a functional analysis of the mCA IX protein focused on pH regulation, cell–cell adhesion, and migration. Finally, we revealed an absence of a soluble extracellular form of mCA IX and provided the first experimental evidence of mCA IX presence in exosomes. In conclusion, though the protein characteristics of hCA IX and mCA IX are highly similar, and the transcription of both genes is predominantly governed by hypoxia, some attributes of transcriptional regulation are specific for either human or mouse and as such, could result in different tissue expression and data interpretation.

## 1. Introduction

Hypoxia, as a hallmark of most solid tumors, is a negative prognostic factor and is associated with an aggressive tumor phenotype and resistance to therapy. Given its foremost role in oncology and obvious impact on prognosis and treatment, precise detection and monitoring of tumor hypoxia are absolutely critical. There are several possibilities/techniques of how tumor hypoxia can be investigated; either directly—using oxygen electrodes, indirectly—utilizing extrinsic (pimonidazole) or intrinsic markers (e.g., carbonic anhydrase IX—CA IX), or by positron emission tomography (in case of a more complex evaluation across an entire tumor volume) [1].

The cellular response to hypoxia is predominantly executed via hypoxia-inducible factor (HIF-1)—a heterodimeric protein consisting of an oxygen-dependent α subunit and a constitutively expressed β subunit. HIF-1, as a transcription factor, upregulates the expression of several tens of genes, including those involved in angiogenesis, erythropoiesis, glucose metabolism, pH regulation, and apoptosis [2]. HIF-1 transcriptional activity is mediated through hypoxia-responsive elements (HREs; 5′-RCGTG-3′) present in the promoters of a wide spectrum of target genes. Primarily due to the unique position of HRE within the *CA9* promoter, CA IX is considered as one of the best endogenous sensors of HIF-1 activity and thus serves as a reliable marker of tumor hypoxia.

CA IX exhibits a distinct expression pattern characterized by limited distribution in normal tissues restricted mainly to the epithelia of the gastrointestinal tract [3]. In contrast, CA IX is very often and strongly expressed in a broad range of tumors where it serves not only as a fundamental pH regulator but also as an essential component of cell migration/invasion machinery [4,5,6,7,8]. Furthermore, CA IX expression is associated with poor prognosis and progression in several types of cancer [9,10,11,12,13]. Therefore, considerable research efforts in recent years have focused on the development, pre-clinical, and clinical evaluation of therapeutic strategies targeting CA IX, either via compounds inhibiting its enzymatic activity or via specific monoclonal antibodies detecting and consequently killing CA IX-expressing cells (reviewed in [14]). The significance of human CA IX (hCA IX) as a promising tumor biomarker and therapeutic target entails an intensive search for relevant biological models. 

Presently, mouse CA IX (mCA IX) appears to be an appropriate candidate for intended preclinical studies. Similar to the human isoform, the highest immunoreactivity for mCA IX was reported in gastric mucosa, while a moderate reaction was detected in the colon and brain [15]. Although both hCA IX and mCA IX seem to possess similar protein characteristics, their transcriptional regulation and protein expression is, at least partially, different and should be taken into account during the evaluation and interpretation of experimental data. Because of these facts, we investigated the transcriptional regulation of the *Car9* gene. In addition, we performed functional analyses of the mCA IX protein and investigated its extracellular forms. Recognition of similarities and differences between the mouse and human CA IX orthologs will allow us to identify a context in which mouse can serve as a suitable model organism for CA IX studies.

## 2. Results

### 2.1. Transcriptional and Post-Transcriptional Regulation of the Car9 Gene

*Car9*, the gene encoding mouse CA IX, was first described and identified almost 20 years ago [16]. In summary, the *Car9* gene covers 1959 base pairs (with the coding region between 31 and 1344) and consists of 11 exons and 10 introns. All exons are small in size except for the first and the last one.

#### 2.1.1. In Silico Analysis of the Mouse Car9 Promoter Sequence

Within the human *CA9* promoter, five protected regions (PRs) were detected; four were identified as activating cis-elements, and the remaining one, PR4, was proven to act as a silencer [17]. Clustal Omega alignment of the human (*CA9*) and mouse (*Car9*) promoter sequences—200 nucleotides upstream and +50 downstream from the transcription start site (TSS)—revealed relatively high identity and similarity between the human and mouse promoters (75.8% for both; Figure 1A), suggesting the existence of the similar regulatory mechanism responsible for *Car9* transcription. In silico analysis of the mouse *Car9* upstream region (206 base pairs) using MatInspector was performed to investigate putative transcription factor (TF) binding sites. The majority of the TFs found via MatInspector analysis of the *Car9* sequence were similar to the human ortholog (e.g., HIF-1, SP1, AP1), and therefore, we decided to analyze their role in the transcriptional regulation of *Car9*.

#### 2.1.2. Hypoxia- and Density-Induced Activity of the Mouse Car9 Promoter

Previously described hypoxia- and density-induced expression of mCA IX in three different mouse cell lines (L-929, MEF-wt, and TSA) was revealed by Western blotting [18]. Here, we employed luciferase reporter analysis and transfected HeLa cells with the pGL3-*Car9* FL (FL = full length) construct (−191/+47). The activity of the *Car9* promoter was investigated at various cell densities (indicated as sparse, medium, or dense) and after incubation in normoxia or hypoxia for 24 h. As clearly shown in Figure 1B, the incubation of transfected cells in hypoxia resulted in elevated *Car9* promoter activity. In addition to hypoxia, the promoter activity of *Car9* was also upregulated by high cell density. Hypoxia-dependent induction was observed under sparse (2-times), medium (4-times; *p* < 0.05), and dense (almost 20 times; *p* < 0.01) conditions. Taken together, the highest induction of the *Car9* promoter was revealed in dense culture cells incubated under hypoxia. To compare the induction between mouse and human promoters, HeLa cells were similarly transfected with the *CA9* promoter construct (pGL3-*CA9*), and transfected cells were cultivated under the same conditions. Luciferase reporter assays revealed that both promoters were induced to comparable levels by hypoxia and density (Figure 1B).

Additionally, the hypoxia-dependent promoter activity of *Car9* was confirmed by point-mutation analysis. The *Car9* promoter construct with mutated HIF-1-binding site (HRE; indicated in bold and underlined; 5′-TGC**TTT**TA-3′) localized −1/+7 around the TSS was generated and transfected into HeLa cells. As expected, disruption of the HIF-1-binding site had a dramatic effect on the *Car9* promoter activity (Figure 1C). The remaining activity of the HREmut construct was only 2.4% of the FL *Car9* promoter.

#### 2.1.3. The role of SP1 and AP1-Binding Sites in the Transcriptional Activity of Car9

Inspection of the *Car9* promoter revealed almost identical nucleotide sequences with the human *CA9* promoter (indicated as PR1-5), suggesting the recognition and binding of similar transcription factors. The core promoter of the human *CA9* gene contains activating cis-elements (Figure 1A), including HIF-1-binding element—HRE (−10/−3), SP1-binding regions (−45/−24 and −163/−145), and AP1-binding regions (−74/−56 and −101/−85) [17,19,20,21]. 

To evaluate the role of putative cis-acting regulatory elements in *Car9* gene expression, we prepared a set of promoter constructs through PCR-based in vitro mutagenesis. With respect to the mutations generated and analyzed within the human *CA9* promoter sequence, we prepared similar luciferase promoter constructs with the same nucleotide mutations. As shown in Figure 1C, the introduction of double-nucleotide mutations into either SP1- (within PR1) or AP1- (within PR2) binding sites led to significant inhibition of *Car9* promoter activity under normoxic, as well as hypoxic conditions. Mutation of the putative AP1-binding site (AP1mut construct) resulted in 56.2% and 60% activity (compared to FL *Car9*) in normoxia and hypoxia, respectively. A more pronounced effect was observed in the case of disrupted SP1-binding site (SP1mut promoter construct), where only 11.6% and 5.62% activity was retained under normoxic and hypoxic conditions, respectively. Taken together, mutation analysis of both putative regulatory elements proved their relevance for the transcriptional activity of *Car9*.

#### 2.1.4. Significant Differences Observed within PR4 and PR5 Regions

Although a high level of identity (almost 76%) was achieved using Clustal Omega for sequence alignment of the human and mouse promoters, a relatively high number of discrepancies could be observed within the PR4 (Figure 1D) and PR5 region (Figure 1A). 

In the case of the human *CA9* promoter, the PR4 region was confirmed to function as a promoter-, position-, and orientation-independent silencer (Figure 1D) [17]. Thus, we were interested in the evaluation of the role of the PR4 region in the activity of the *Car9* promoter. A mouse deletion construct lacking the PR4 region (*Car9* ∆PR4) was prepared similarly to the human *CA9* ∆PR4 construct by inverse PCR, and its activity was analyzed by luciferase reporter assay. Surprisingly, the deletion of PR4 from the mouse promoter construct led to a decreased activity under both normoxic (41.6%) and hypoxic (45.6%; *p* < 0.01) conditions (Figure 1E). These results are in contrast to the human *CA9* ∆PR4 construct (as shown in Figure 1E), where the abrogation of PR4 resulted in a significantly increased *CA9* promoter activity (more than 5-fold under both conditions). Therefore, the PR4 region was not proven to act as a silencer but seems to play a positive role in the transcriptional regulation of the mouse promoter.

In addition to PR4, an inspection of the mouse promoter sequence within the PR5 element revealed relatively low homology with the human counterpart (Figure 1A). In the case of the human promoter, the PR5 sequence possesses a significant identity with PR1. Through supershift EMSA (Electrophoretic Mobility Shift Assay), this region was identified as a second SP1/SP3-binding site [20]. Interestingly, this putative sequence identified as a SP1/SP3-binding site is not preserved within the mouse promoter.

### 2.2. Characterization of the Mouse CA IX Protein

Although considerable effort has been directed towards the description of both the distribution, as well as the expression of mouse CA IX within different organs and tissues, only limited data regarding its function are available. 

The *Car9* cDNA has a coding capacity for a 437-amino acid protein with a molecular weight of 47.3 kilodaltons [16]. According to the amino acid sequence alignment (using Clustal Omega), the mouse protein shows 68.4% identity with its human homolog, and moreover, it has a similar predicted domain composition (Figure 2A). Interestingly, the highest sequence similarity is present within the catalytic (CA) domain, while most of the sequence differences are observed in the proteoglycan (PG)-like region.

#### 2.2.1. Functional Analyses of mCA IX

The extracellular catalytic domain of hCA IX significantly contributes to pH regulation and thereby is essential for the adaptation of tumor cells to the physiological stress of hypoxia and acidosis. Based on the main hCA IX protein function, we first investigated the effect of mCA IX expression on acidification of extracellular pH under hypoxia. Extracellular pH (pHe) was measured in MDCK cells transfected with either mCA IX (pSG5C-*Car9*) or empty vector and incubated in normoxia or hypoxia for 48 h. As shown in Figure 2B, measurements revealed the same capability of mCA IX to acidify pHe under hypoxia as had been described for human CA IX [4].

In addition to pH regulation, CA IX expression decreases cell–cell adhesion and increases cell migration [6,22]. Similar to the previous experiments with pHe measurements, we used MDCK cells to examine the influence of ectopically expressed mCA IX on cell dissociation. Results from the dissociation assay clearly show that mCA IX expression led to a higher dissociation capacity of transfected MDCK cells (Figure 2C). MDCK-mCA IX cells were dissociated to a significantly higher degree than control cells transfected with an empty vector (MDCK-mock; *p* < 0.001). Moreover, the contribution of mCA IX expression to increased dissociation of MDCK cells is comparable to the effect of hCA IX expression in MDCK cells (MDCK-hCA IX). The expression, as well as correct membrane localization of both proteins transfected into MDCK cells, was validated through immunofluorescence staining (Figure 2D).

Following the confirmation of reduced adhesion capacity of mCA IX-expressing cells, we next examined the participation of mCA IX in cell migration using the xCELLigence real-time cell analyzer. For this analysis, we decided to employ murine fibroblasts expressing mCA IX either naturally (L-929 cells) or ectopically (NIH3T3 cells). Thus, migration of L-929 cells pre-incubated in hypoxia for 48 h (for the highest induction of mCA IX expression) was compared to normoxic L-929 cells and simultaneously, pSG5C-*Car9*-transfected NIH3T3 cells (NIH3T3-mCA IX) to mock-transfected NIH3T3 cells (NIH3T3 mock). As shown in Figure 2E, mCA IX-expressing L-929 cells showed increased migration compared to their normoxic counterparts. A similar effect was observed in the case of NIH3T3 cells expressing mCA IX. Taken together, all experiments described above pointed to analogous functions of both human and mouse CA IX.

#### 2.2.2. Extracellular Forms of mCA IX—mCA IX Ectodomain Shedding

CA IX as a transmembrane protein can also be detected as a soluble metalloprotease-cleaved form or as a soluble membrane-bound form released on cell-derived exosomes [23,24,25,26]. We were, therefore, interested in identifying and characterizing extracellular forms of mCA IX, because of all the above-mentioned functional similarities identified between both orthologs.

The extracellular domain (ECD) of human CA IX (consisting of a large CA domain which is N-terminally linked to a PG domain) is connected to a short intracytoplasmic (IC)-tail via a transmembrane (TM) region and can be released into the cell culture medium or body fluids [23,27]. Previously described cleavage of the CA IX ECD mediated by TNF-α-converting enzyme TACE/ADAM17 [24], together with the fact that 90% identity and more than 94% similarity in the amino acid sequence of TACE/ADAM17 exists between both species, prompted us to investigate the shedding of mCA IX. Like in the migration assay, we employed murine fibroblasts NIH3T3 and L-929. To accelerate the shedding of mCA IX, both cell types were treated with phorbol-12-myristate-13-acetate (PMA) for 3 h. Following acetone precipitation, the level of mCA IX present in the culture medium, as well as in the corresponding cell extracts, was assessed by Western blotting. Even though shedding was induced with PMA, and the expression of TACE/ADAM17 was confirmed in L-929 cells under normoxic as well as hypoxic conditions (data not shown), mCA IX was not present in the culture media from any of the examined cells and was detected only in the cell extracts (Figure 3A). In contrast, the presence of hCA IX ectodomain in culture medium samples and the induction of hCA IX shedding after PMA treatment were observed in human BT-20 cells derived from breast carcinoma. 

To investigate the absence of mCA IX release from murine cells, we performed an inspection of the amino acid sequence covering the segment between the CA and TM domains of both species. As shown in Figure 3B, sequence alignment revealed only 54% similarity between mCA IX and hCA IX, and moreover, the absence of three amino acids in the sequence where TACE/ADAM17 could act and cleave the protein. 

#### 2.2.3. Extracellular Forms of mCA IX—Presence of CA IX in Exosomes Derived from Murine Cells

Although the release of mCA IX in the form of an ectodomain was not confirmed, we continued the investigation of mCA IX in exosomes. For the isolation of exosomes, we employed murine cell lines naturally expressing mCA IX. Exosomes isolated from normoxic, as well as hypoxic melanoma B16-F0 cells, were characterized and quantified by Nanoparticle tracking analysis (NTA). NTA was carried out using a NanoSight NS500 (5 × 60 s run) confirming the size distribution with a mean = 102 nm and an average mode = 52 for normoxic exosomes (Figure 3C). Exosomes derived from B16-F0 cells incubated under hypoxic conditions for 48 h showed relatively similar dimensions with a mean = 95 nm and an average mode = 49. However, the overall concentration of hypoxic exosomes was higher (8.43 × 10^12^ versus 7.34 × 10^12^ of normoxic exosomes/mL). The expression of mCA IX in exosomes isolated from murine B16-F0 cells was determined using ELISA. As shown in Figure 3D, the level of mCA IX in exosomes was increased more than 1.5 times in response to hypoxia. 

Similar to the case of B16-F0 exosomes, we confirmed the presence of mCA IX in exosomes derived from L-929 cells using Western blotting analysis (Figure 3E). Moreover, higher expression of mCA IX was observed in cell lysates, as well as in exosomes isolated from L-929 cells incubated under hypoxic conditions. The correct isolation of exosomes was verified through the presence of a specific marker of exosomes—CD63. 

### 2.3. Expression of mCA IX

To summarize all available data regarding the expression of mCA IX in different tissues and cell types, we utilized the Genevestigator platform representing a tool that integrates high-quality public microarray data. Figure 4 shows results from Genevestigator analysis of *Car9* according to tissue and cell type (A) and across different stages of development (B). 

## 3. Discussion

Even though the expression and distribution of mouse CA IX have been quite well established, relatively little is known about the molecular mechanisms controlling *Car9* expression, as well as its protein function.

Hypoxia- and density-induced expression of mouse CA IX, together with similar intratumoral distribution, has already been described using newly generated monoclonal antibodies [18]. Although the introduction of novel monoclonal antibodies has enlarged the possibilities of studying mCA IX expression, no relevant data on transcriptional regulation and *Car9* promoter have been published so far. In the present study, we focused on expanding our knowledge about the transcriptional regulation of gene coding for mouse carbonic anhydrase IX. Through in silico analysis of the mouse *Car9* upstream region, relatively high sequence identity and similarity to human *CA9* was revealed. Moreover, several putative binding sites for transcription factors (TFs) were identified. As expected, the same TFs responsible for human *CA9* transcriptional activation, such as HIF-1, SP1, and AP1, which bind regions marked as HRE, PR1, and PR2, respectively, were also recognized to bind within the mouse *Car9* sequence. Although these regions were not identified as protected via DNase I footprinting, they are identical (HRE) or almost identical (PR1 and PR2) to the *CA9* promoter [17]. In this study, using promoter constructs with point mutations in either HIF-1-, SP1- or AP1-binding sites, all three TFs were confirmed to play an important role in *Car9* transcription. Analogously to the *CA9* promoter, the presence of the HIF-1-binding element (within mHRE) and the SP1-binding element (within mPR1) is absolutely essential, whereas AP1 (via mPR2 binding) only enhances *Car9* promoter activity. 

In addition to HIF-1, SP1, and AP1, we were also interested in validating the rest of the *Car9* promoter sequence. To investigate the role of the PR4 region in *Car9* transcription, we prepared a PR4 deletion mutant (*Car9* ∆PR4) from the pGL3-*Car9* FL promoter construct by inverse PCR. Unlike human PR4, which has been shown to be a silencer [17], the elimination of PR4 within *Car9* resulted in a significant decrease in promoter activity. Therefore, PR4 exhibits a positive effect on *Car9* transcription. Importantly, an inspection of the PR4 sequence within the mouse promoter revealed 88.5% identity as well as similarity to the human PR4 sequence. However, the direct repeat AGGGCacAGGGC sequence within the *CA9* promoter required for efficient repressor binding (as described by Kaluz and colleagues [17]), is not preserved within the *Car9* promoter and thus, could not be recognized by a yet uncharacterized transcriptional factor/repressor. According to the significant downregulation of the *CA9* mRNA expression, microorchidia 2 (MORC2) was suspected as a potential repressor [28]. Shao and colleagues revealed that the PR4 region was important for the repressor function, and moreover, MORC2 and the histone deacetylase 4 (HDAC4) protein complex were assembled on the PR4 region within the *CA9* promoter. They confirmed that both protein binding sites were simultaneously present in the PR4 sequence. However, they did not establish consensus binding sites for either of these two proteins [28]. Although the role of the aforementioned proteins in decreased acetylation of the *CA9* promoter seems to be important, the relevance for *CA9* transcription remains to be better elucidated. 

In addition to the PR4 region, the PR5 sequence is also considerably different and only partially overlaps with the human *CA9* promoter (as shown in Figure 1A). Investigation of the PR5 sequence within the *CA9* promoter revealed significant homology to PR1, and moreover, PR5 was confirmed as another SP1/SP3-binding site (functionally equivalent to PR1) with a positive effect on the transcriptional activity of *CA9* [20]. Interestingly, the putative SP1-binding sequence is not preserved within the PR5 region of the *Car9* promoter, and thus, we do not expect the binding of the SP1 transcription factor. However, the absence of repressor binding within the PR4 region, as well as the fact that the remaining promoter sequence could be recognized and occupied by other TFs, suggests that the regulation of *Car9* transcription is species-specific. For instance, in silico analysis of the *Car9* promoter sequence revealed a putative TCF/LEF1-binding site (5′-CTTTGAT-3′, involved in the Wnt signal transduction pathway) [29] localized at the beginning of the PR5 sequence, which is not present within the *CA9* promoter. In conclusion, our results indicate that the major effect on *Car9* transcription is executed via HIF-1, SP1, and AP1 TFs, just like in the case of human *CA9*. The rest of the *Car9* promoter sequence seems to have a minor effect on transcription. However, in the absence of hypoxia, *Car9* transcription could be controlled by a different mechanism, e.g., via TFs that recognize the rest of the promoter sequence and affect its transcription. 

Alternative splicing (AS) as a mechanism of post-transcriptional regulation of mouse CA IX represents another aspect similar to the human isoform. In accord with the human AS variant, the mouse AS variant lacking the C-terminal part of the CA domain (exons 7-8) is localized in the cytosol and is expressed independently of hypoxia [30].

Expression of the CA IX protein in mouse tissues has been intensively studied earlier [15,16,31,32,33,34,35,36,37]. First, the gene encoding the mouse CA IX was identified and cloned [16]. Furthermore, the generation of CA IX-deficient mice allowed the elucidation of the physiological relevance of mCA IX to be studied. Gut and colleagues revealed the phenotypic consequences of CA IX deficiency (gastric hyperplasia of the glandular epithelium with many cysts), suggesting an important role of CA IX in morphogenesis and homeostasis of the gastric epithelium through the control of cell proliferation and differentiation. To better understand the hyperplastic phenotype of the stomach mucosa of *Car9*^−^/^−^ mice, a genome-wide expression analysis was performed by Kallio and colleagues [38]. As expected, microarray analysis revealed several genes involved in developmental processes and cell differentiation. Interestingly, some of the affected genes were involved in digestion, as well as in the function of the immune and defense responses. The impact of the genetic ablation of *Car9* on gastric mucosa was also investigated by Li and co-workers [37]. Lack of mCA IX resulted in persistent acid backflux via claudin-18 downregulation, causing the loss of parietal cells, hypergastrinemia, and foveolar hyperplasia. 

Following the first mention of mouse CA IX, Hilvo and colleagues [15] used RT PCR, Western blotting, and immunohistochemistry (IHC) to describe a similar expression pattern of the mouse, human, and rat proteins (although some differences existed, especially in the gut epithelium). Surprisingly, strong expression of the *Car9* mRNA was revealed in the kidney and skeletal muscle, while no positive signal was observed at the protein level by WB and IHC. Interestingly, a weak positive signal for CA IX in the kidney was also revealed during mouse embryonic development [33]. In general, a relatively wide distribution of mCA IX within several mouse tissues was described during embryonic development, although the signal intensity was low or moderate. On the other hand, the youngest investigated mouse embryos (at a gastrulation stage) were completely CA IX-negative [33]. 

Both previously mentioned studies have definitely extended our knowledge of the mCA IX expression pattern in embryos, as well as in several adult mouse tissues. However, we are aware that in silico Genevestigator analysis of the *Car9* mRNA expression presented in Figure 4A,B only partially overlaps with mCA IX protein expression as described by Hilvo and Kallio [15,33]. A possible explanation of this discrepancy could be the primary antibody used for mCA IX detection. All the previously mentioned studies employed rabbit polyclonal serum raised against mCA IX, which was probably less sensitive or led to the detection of non-specific signal due to its polyclonal character in comparison to our monoclonal antibody AM4-3 [18]. 

Tumor hypoxia is of paramount importance for CA IX because of the regulation of its expression, as well as its enzymatic activity. Numerous studies have confirmed the prominent role of human CA IX in pH regulation, which promotes tumor cell survival [4,5,39]. Moreover, recent studies have also shown the significance of CA IX expression for tumor cell migration and invasion [6,7,40,41]. To validate the existence of the same protein characteristics, we performed a functional analysis of mCA IX. With respect to the amino acid sequence, the mouse protein showed more than 68.4% identity and almost 75% similarity to its human homolog and had a similar predicted domain arrangement. The highest discrepancies between mCA IX and hCA IX could be observed within the PG region [16]. Therefore, it is not surprising that the most commonly used monoclonal antibody, M75, directed to the N-terminal proteoglycan domain of hCA IX, is not able to recognize the mouse protein. However, high sequence similarity within the CA domain, as well as the presence of key histidine residues in the enzyme’s active site, suggested the preservation of the catalytic activity of mCA IX. Using MDCK cells, we confirmed that mCA IX was able to acidify extracellular pH under hypoxic conditions, and moreover, its enzymatic activity was comparable to the activity of the human ortholog described by Svastova and colleagues [4].

Importantly, the ability of human CA IX to mediate extracellular acidification in the hypoxic cell is dependent on protein kinase A (PKA) activity. Ditte and co-workers revealed that Thr443 phosphorylation at the intracellular domain of hCA IX is critical for its activation in hypoxic cells, with the fullest activity of CA IX also requiring dephosphorylation of Ser448 [42]. Simko and colleagues reported that elevated intracellular concentrations of cAMP can enhance PKA activity and moreover, such an elevation of cAMP levels is mediated through adenylyl cyclase 6 and 7 and their transcriptional activation in hypoxia [43]. In addition, the IC tail of hCA IX contains another phosphorylation site, Tyr449, which occurs in kidney cancer cells and is linked to downstream activation of the AKT/PI3K pathway [44]. When using the Scansite motif scanner, we were not able to identify a PKA canonical phosphorylation-site motif (R-X-X-S/T) within the IC amino acid sequence of mCA IX. Interestingly, the best score was achieved by AKT1 kinase, which potentially could phosphorylate the Thr421 in the IC tail of mCA IX (similar as to how Thr443 is phosphorylated by PKA in hCA IX). However, the AKT1 phosphorylation-site motif was only predicted, and further studies are required to determine a proposed role of AKT1 in phosphorylation of the intracellular tail of mCA IX. 

In addition to pH regulation, we also investigated and confirmed the contribution of mCA IX expression to decreased cell–cell adhesion and increased migration. Both aforementioned phenomena regarding human CA IX were first described in our laboratory [4,6], and, given their significance, they highlighted the relevance of CA IX. Since we have now confirmed the same protein functions for mCA IX, the use of the mouse model is recommended for further studies of CA IX’s role in tumor development and for its preclinical investigation.

As part of our ongoing effort to understand the mCA IX protein, we have also investigated the existence of its extracellular forms. Shedding is an important regulatory mechanism affecting the abundance of membrane-bound molecules and significantly impacts the biological functions of these proteins by converting them into soluble molecules that can either be biologically active variants or inactive decoys. In the case of human CA IX, Zatovicova and colleagues showed that metalloprotease TACE/ADAM17 was responsible for ECD cleavage [24]. Therefore, we were interested in the detection of the soluble mCA IX ectodomain in cell culture media from murine cells. Even though we examined and confirmed the presence of TACE/ADAM17 in the tested cells, the ectodomain was not revealed in the cell culture media from either NIH3T3-mCA IX or L-929 cells. Moreover, PMA treatment did not affect the release of mCA IX ECD in any of the examined cells. Further investigation revealed that despite a very high amino acid similarity, the stalk region between the CA and TM domains of both species differed substantially. In humans, the generation of constructs with deletions within the stalk region covering amino acids 392–414 enabled the identification of a non-shed CA IX mutant (Kajanova and Zatovicova et al., manuscript submitted). The deletion of ten amino acids within the stalk region of hCA IX completely abolished basal, as well as activated shedding, and additionally led to the identification of a unique cleavage site. It is not surprising that the absence of three amino acids within the putative cleavage site (as shown in Figure 3B) renders the shedding of mouse ectodomain impossible. 

Since human CA IX has been detected in exosomes isolated from renal and prostate cancer cell lines [25,26], we hypothesized that exosomes expressing mCA IX could be released from murine B16-F0 and L-929 cell lines. Western blot analysis, as well as ELISA, enabled us to reveal mCA IX protein expression in exosomes isolated from both cell lines. In addition, consistent with the intracellular expression level, the expression of mCA IX was evidently higher in exosomes isolated from B16-F0 and L-929 cells grown in hypoxia for 48 h. Similarly, Horie and colleagues found that CA IX was elevated in exosomes isolated from several renal carcinoma cell lines following hypoxia or treatment with CoCl_2_ (a hypoxia mimic agent) [25]. Interestingly, we observed that hypoxia also affected the production of exosomes, and thus, the concentration of exosomes isolated from hypoxic murine cells was higher than in the normoxic counterparts. Hypoxia-induced exosome release has been shown to be HIF-dependent [45], although specific downstream target genes required for exosome release have not been identified. Recently, Zhang and colleagues described significantly increased exosome production from rat kidney tissue (renal tubular cells) in hypoxia [46].

Based on all the previously mentioned facts, we can conclude that the mouse CA IX possesses identical protein functions as the human ortholog. However, we were not able to identify its soluble metalloprotease-cleaved form, presumably due to the lack of a TACE/ADAM17 cleavage site within the stalk region between the CA and TM domains of mCA IX. Therefore, the only extracellular form of mCA IX is released in exosomes and, similar to hCA IX, in a hypoxia-dependent manner. Finally, the mouse CA IX protein characteristics, as well as all the available facts regarding the *Car9* gene, are summarized in Figure 5. 

## 4. Materials and Methods 

### 4.1. Cell Cultures

Canine MDCK epithelial cells (ATCC CCL-34), murine embryonic fibroblasts NIH3T3 (ATCC CRL-1658), murine fibroblasts derived from subcutaneous connective tissue L-929 (ATCC CCL-1), murine melanoma cell line B16-F0 (ATCC CRL-6322), human tumor cell lines BT-20 (ATCC HTB-19) derived from breast carcinoma, and HeLa (ATCC CCL-2) derived from cervical carcinoma were cultivated in DMEM supplemented with 10% FCS (BioWhittaker, Verviers, Belgium) in a humidified atmosphere with 5% CO_2_ at 37 °C. Hypoxic treatments were performed in an anaerobic workstation (Ruskin Technologies, Bridgend, UK) in 2% O_2_, 5% CO_2_, 10% H_2,_ and 83% N_2_ at 37 °C.

### 4.2. Plasmids 

The *Car9* promoter construct was generated by insertion of −191/+47 *Car9* genomic region amplified by PCR upstream of the firefly luciferase gene into pGL3-Basic luciferase reporter vector (Promega, Madison WI, USA). The generation of human *CA9* promoter construct pGL3-*CA9* was described previously [47]. pRL-TK *Renilla* vector served for the control of transfection efficiency. 

HRE-, SP1-, and AP1-binding sites in the pGL3-*Car9* promoter construct were mutated using certain mutagenic oligonucleotides and their reverse counterparts (summarized in Table 1) through PCR-based in vitro mutagenesis. The mutated fragments were amplified in the second round of PCR using pGL3-basic specific primers, digested with *Sac*I and *Xho*I, and inserted into pGL3-basic. A deletion construct lacking the PR4 region was prepared by the amplification of the construct containing −191/+47 fragment in pGL3 by inverse PCR, using sense and antisense primers from the downstream and upstream PRs, respectively. The resulting PCR products were gel-purified, phosphorylated, and ligated. The resulting mutations and deletions were verified by sequencing.

The eukaryotic expression plasmids pSG5C-*Car9* and pSG5C-*CA9* were described previously [18,48]. The generation of mCA IX-expressing NIH3T3 and MDCK cells was described previously [18].

### 4.3. Transient Transfection and Luciferase Assay

HeLa cells were plated into 35-mm Petri dishes to reach an approximately 70% monolayer density the following day. Transient transfection was performed with 2 μg of promoter-containing luciferase construct pGL3-*Car9* and 100 ng of pRL-TK plasmid using Turbofect reagent (Thermo Fisher Scientific, Waltham, MA, USA) according to the manufacturer’s recommendations. Cells transfected with empty vectors served as negative controls. One day later, transfected cells were trypsinized and then plated in triplicate on 24-well plates. Transfected cells were allowed to attach overnight and were then transferred to hypoxia for an additional 24 h. Reporter gene expression was assessed using the Dual-Luciferase Reporter Assay System (Promega), and the luciferase activity was normalized against Renilla activity. 

### 4.4. Immunofluorescence

Cells grown on glass coverslips were fixed in ice-cold methanol at −20 °C for 5 min. Non-specific binding was blocked by incubation with PBS containing 1% BSA for 30 min at 37 °C. Then the cells were incubated with primary antibodies (AM4-3 for mCA IX and M75 for hCA IX detection; [18,49]) diluted in PBS with 0.5% BSA (PBS-BSA) for 1 h at 37 °C, washed three times with PBS-BSA for 10 min, incubated with anti-mouse FITC-conjugated horse antibody (Vector Laboratories, Burlingame, CA, USA) diluted 1:300 in PBS-BSA for 1 h at 37 °C and washed as before. Finally, the cells were mounted onto slides in the Fluorescent Mounting Media (Calbiochem, Darmstadt, Germany), analyzed with Leica DM4500B microscope and photographed with a Leica DFC480 camera. 

### 4.5. Cell Dissociation Assay

MDCK cells were grown for 3 days to form a highly dense monolayer. After washing twice with PBS containing 2 mM CaCl_2_ and 2 mM MgCl_2_, the cells were detached using a cell scraper, passed 30 times through the Pasteur pipette, and counted using a Coulter Counter (Beckman Coulter, Brea, CA, USA). The extent of dissociation was expressed as a ratio of Np/Nc, i.e., the number of disrupted particles per total number of cells obtained by counting the cells from the parallel monolayer fully dissociated in PBS.

### 4.6. Real-Time Monitoring of Migration with the xCELLigence System

The xCELLigence cell index impedance measurements were performed using the CIM-Plate16 placed in the RTCA DP station according to the instructions of the supplier (Roche, Basel, Switzerland). Both NIH3T3 (mCA IX- and mock-transfected) as well as L-929 (normoxic and hypoxic) cells were trypsinized, resuspended at the density of 400,000 cell/mL in serum-free medium, added to the top chamber of the CIM-Plate, and allowed to migrate towards bottom chamber containing medium with 10% FCS as a chemoattractant. The CIM-Plate 16 was placed in the RTCA DP station, and migration was monitored every 15 min for 100 h. Migration was expressed as a cell index representing relative change impedance monitored every 15 min for 10 h.

### 4.7. Isolation of Exosomes

Exosomes were isolated from 2 × 10^7^ of B16-F0 cells cultivated in 10% FCS DMEM for the first 48 h, followed by a 48 h cultivation in FCS/ATB-free medium under normoxic, as well as hypoxic conditions. Then, the medium was centrifuged twice at 300× *g* for 5 min and 1300× *g* for 10 min to remove cell debris, and filtered through a 0.22 μm filter (Merck Millipore, USA). The pre-cleared medium was concentrated using 100 kDa MWCO Amicon Ultra Centrifugal Filter (Merck Millipore, USA) at 3000× *g*. Finally, exosomes were isolated using a Total exosome isolation kit (Invitrogen/Thermo Fisher Scientific, USA) during an overnight incubation at 4 °C followed by centrifugation at 10,000× *g* for 1 h. The exosome pellet was resuspended in PBS.

### 4.8. Nanoparticle Tracking Analysis

For the rapid in vitro measurements of exosomes, we performed Nanoparticle tracking analysis (NTA) using NanoSight NS500 equipped with an sCMOS Trigger camera and a 405 nm laser (Malvern Instruments Ltd., Malvern, UK). NTA utilizes the properties of both light scattering and Brownian motion to obtain the size distribution and concentration measurements of particles in liquid suspension. The measured data were analyzed using NTA2.3 analytical software. Each sample was diluted in PBS before the measurements to optimize the number of particles. Samples were measured in quintuplicate in 60-s videos with manual shutter and gain adjustments.

### 4.9. ELISA

The expression of mCA IX in exosomes isolated from B16-F0 cells was determined using ELISA. Protein concentrations were determined using the BCA protein assay reagent (Pierce, Rockford, IL, USA). Fifty micrograms/well of isolated exosomes was coated on the surface of microplate wells overnight at 37 °C. Primary mouse monoclonal antibody AM4-3 (undiluted medium) was added and incubated for 2 h at RT. Peroxidase-labeled swine anti-mouse IgG (diluted 1:5000; Sigma–Aldrich, USA) was used as the secondary antibody. Results are expressed as O.D. values of absorbance measured at 492 nm.

### 4.10. Western Blotting Analysis

L-929 cells incubated for 48 h in normoxia or hypoxia were solubilized in ice-cold RIPA buffer (1% Triton X-100; 1% deoxycholate, 150 mM NaCl, pH 7.2) containing inhibitors of proteases (Roche Applied Science, Mannheim, Germany) for 30 min on ice. Cell lysates were collected, cleared by centrifugation at 10,000× *g* for 10 min at 4 °C. Protein concentration was determined using the BCA protein assay reagent (Pierce, Rockford, IL, USA). Total protein extracts were separated in 10% SDS-PAGE and transferred onto PVDF membrane (Immobilon^TM^-P, Milliopore, Billerica, MA, USA). For mCA IX detection, the membrane was incubated with mouse monoclonal AM4-3 antibody (10 μg/mL) diluted in blocking buffer for 2 h. Secondary anti-mouse peroxidase-conjugated antibody (Sigma–Aldrich, St. Louis, MI, USA) was diluted 1:5000 in blocking buffer. For CD63 analysis, mouse monoclonal antibody (1 μg/mL, ThermoFisher Scientific, Waltham, MA, USA) was diluted in 3% BSA in PBS with NP-40. For loading control, the membrane was probed with mouse monoclonal anti-actin antibody (Cell Signaling, Danvers, MA, USA). After treatment, the membrane was washed and developed by enhanced chemiluminescence using an ECL kit (Amersham Pharmacia Biotech, Buckinghamshire, UK). 

### 4.11. Shedding

L-929, NIH3T3, NIH3T3-mCA IX, and BT-20 (50,000 cells/cm^2^) were seeded into DMEM with 10% FCS and incubated in normoxic or hypoxic conditions for 48 h. PMA (phorbol-12-myristate-13-acetate, final concentration 20 µM) treatment was performed at the end of incubation for 3 h. Control cells were treated with DMSO. RIPA lysates (prepared as described above) and conditioned media were harvested after the PMA treatment and centrifuged at 400× *g* for 5 min. Supernatants were then mixed with ice-cold acetone (1:4) and incubated overnight at −20 °C. After centrifugation, all pellets were resuspended in RIPA lysis buffer. Western blotting analysis was performed using either AM4-3 (10 µg/mL in 5% non-fat dry milk with 0.2% Nonidet P40 in PBS) or M75 (hybridoma medium diluted 1:3 in 5% non-fat dry milk with 0.2% Nonidet P40 in PBS) for 1 h. Secondary anti-mouse-HRP (Horseradish Peroxidase), as well as an anti-actin antibody, was used similarly as described above. 

### 4.12. Bioinformatics 

In silico analysis of the *Car9* promoter was performed using the MatInspector program (https://www.genomatix.de; [50,51]). The promoter sequence was extracted directly from the ElDorado genome database after the *Car9* gene submission. The accurate position of predicted binding elements was calculated according to the transcription start site (TSS). Sequence alignments (nucleotides as well as amino acids) were performed using the Clustal Omega sequence alignment program (https://www.ebi.ac.uk/Tools/msa/clustalo/; [52]). *Car9* distribution within mouse tissues was analyzed using Genevestigator (https://genevestigator.com; [53]). A Scansite motif scanner (http://scansite.mit.edu; [54]) was used to identify a PKA canonical phosphorylation-site motif.

### 4.13. Statistical Analysis

Results were analyzed by a two-tailed unpaired *t*-test (Student’s *t*-test), and *p* < 0.05 was considered significant.

## 5. Conclusions

All the results obtained in this study provide a complex overview of the mouse CA IX transcriptional regulation, expression, and protein function. In general, there were many similarities between mouse and human gene regulation. In agreement with human *CA9*, the most pronounced effect on *Car9* transcription was observed within the HIF-1- and SP1-binding sites. Point mutations of either of these TF binding sites dramatically decreased the promoter activity. Additionally, the mCA IX protein proved to act in pH regulation, cell–cell dissociation, and migration. All of these facts validate the mouse model in CA IX tumor biomarker studies. However, we also uncovered some aspects of gene regulation and protein expression that are not shared between the species, i.e., the different function of PR4 in the *Car9* promoter and the absence of mCA IX shedding. Understanding these differences could enhance the value of mCA IX in preclinical models.

## Figures and Tables

**Figure 1 ijms-21-00246-f001:**
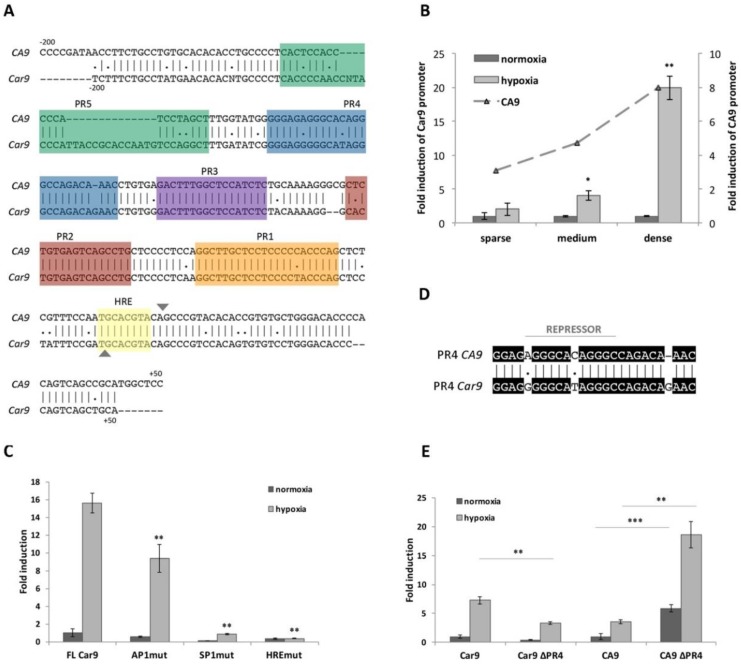
Evaluation of *Car9* promoter activity under different conditions. (**A**) In silico analysis and comparison of the nucleotide sequence (from −200 to +50) of human (*CA9*) and mouse (*Car9*) promoters. Putative protected regions (PRs) and hypoxia-responsive elements (HRE) are highlighted in colors. The transcription start site is marked as▼. (**B**) Reporter gene analysis of pGL3-*Car9* full length (FL) construct (−191/+47) transfected into HeLa cells. Transfected cells were plated in sparse (10,000 cells/cm^2^), medium (40,000 cells/cm^2^), and dense (80,000 cells/cm^2^) cultures. At the same time, the activity of the human pGL3-*CA9* promoter was analyzed and is expressed as a fold induction (ratio of promoter activity in hypoxia to normoxia). (**C**) Comparison of pGL3-*Car9* FL construct with pGL3-*Car9* constructs with mutated binding sites for selected transcription factors: AP1 (AP1mut), SP1 (SP1mut), and HIF-1 (HREmut). (**D**) Alignment of human (*CA9*) and mouse (*Car9*) PR4 sequences. The repressor-binging sequence within the *CA9* promoter is marked with a line. (**E**) Deletion of the PR4 sequence from either mouse (*Car9* ∆PR4) or human (*CA9* ∆PR4) promoter and the impact on the reporter activity. (B–D) 24 h after incubation in either normoxic or hypoxic conditions, transfected cells were lysed and analyzed for promoter activity by the Dual-Luciferase Reporter Assay System. Luciferase activity was normalized against *Renilla* activity. Hypoxic values are expressed as fold induction of the normoxic ones. Data are expressed as means ± SD (error bars), *n* = 3 experiments, * *p* < 0.05, ** *p* < 0.01, *** *p* < 0.001.

**Figure 2 ijms-21-00246-f002:**
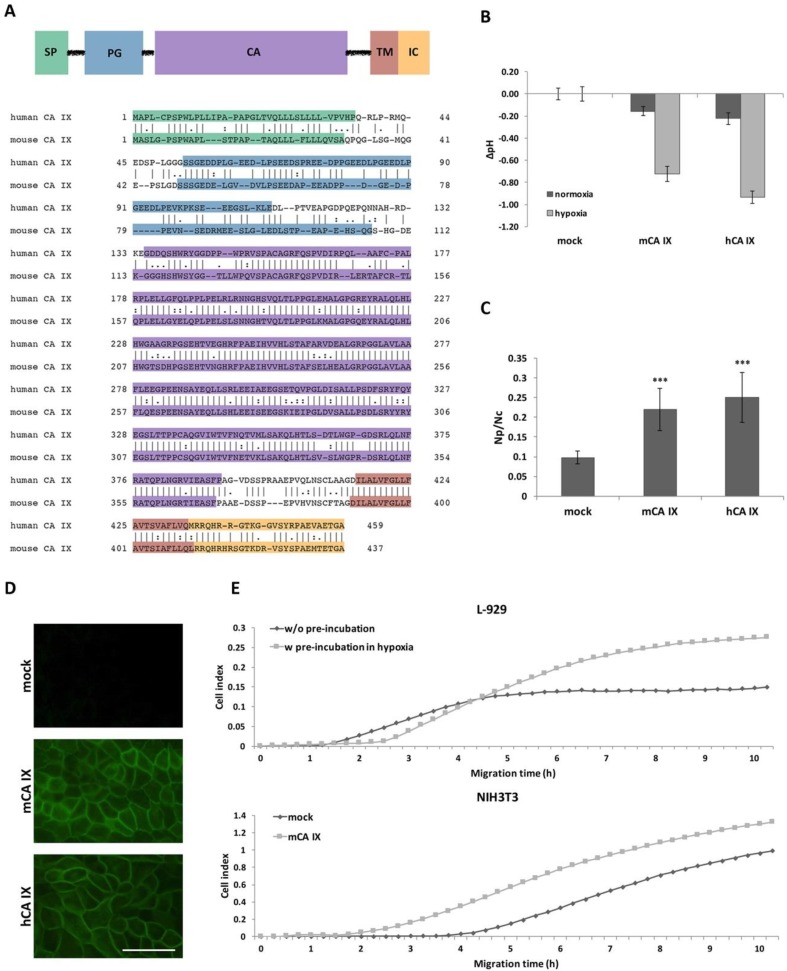
Functional analysis of mouse carbonic anhydrase (mCA IX). (**A**) Domain composition of the CA IX protein relative to the human and mouse CA IX amino acid sequence portrayed below. SP—signal peptide, PG—proteoglycan-like domain, CA—catalytic domain, TM—transmembrane region, IC—intracellular tail. (**B**) mCA IX-mediated acidification of the extracellular pH in hypoxia. Extracellular pH was measured in MDCK cells transfected with either mouse CA IX (MDCK-mCA IX) or human CA IX (MDCK-hCA IX). Control cells were transfected with an empty vector (MDCK-mock). Transfected cells were maintained in normoxia or exposed to hypoxia for 48 h. The graph shows differences in pHe values (∆pH) ± SD from three independent experiments. (**C**) Analysis of the impact of ectopic expression of mCA IX on cell–cell adhesion via dissociation assay. Analogous to hCA IX-expressing MDCK cells, mCA IX-expressing MDCK cells displayed a significantly higher level of dissociation than MDCK-mock cell (*p* < 0.001). The degree of dissociation was calculated as a ratio between the number of dissociated particles (Np) and number of cells (Nc). Mock-transfected MDCK cells served as a negative control. Data are expressed as means ± SD (error bars), *n* = 3 experiments, * *p* < 0.05, ** *p* < 0.01, *** *p* < 0.001. (**D**) Immunofluorescence of MDCK cells transfected with either mCA IX or hCA IX revealed CA IX-specific plasma membrane staining. Mock transfected cells served as a negative control. Scale bar 50 μm. (**E**) Effect of either natural (L-929 cells pre-incubated in hypoxia) or ectopic (NIH3T3 cells transfected with pSG5C-*Car9*) mCA IX expression on cell migration analyzed by xCELLigence real-time cell analyzer. Cells were added in quadruplicates to the upper chambers of the CIM-plate. Migration is expressed as cell index representing relative change in impedance monitored every 15 min for 10 h.

**Figure 3 ijms-21-00246-f003:**
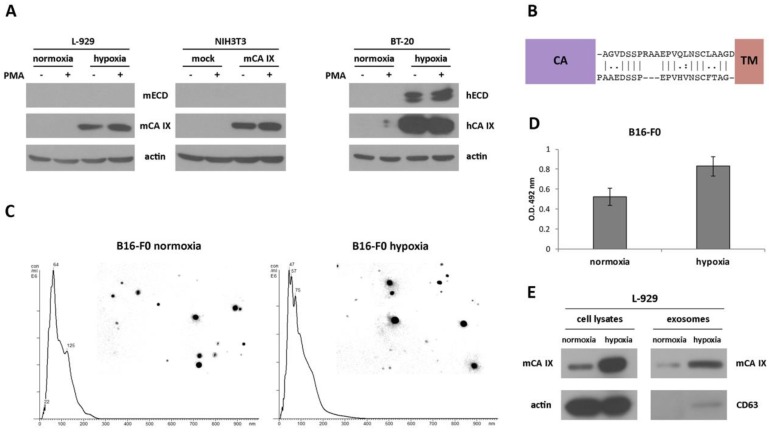
Identification and characterization of the extracellular forms of mCA IX. (**A**) Western blotting analysis of ectodomain (ECD) release from murine L-929 (normoxic and hypoxic) as well as NIH3T3 (mock- and mCA IX-transfected) cells. To accelerate the shedding of mCA IX, both cell types were treated with phorbol-12-myristate-13-acetate (PMA) for 3 h. mCA IX was immunoprecipitated only from cell extracts. BT-20 cells derived from breast carcinoma served as a control for human ECD release detection after treatment with PMA. (**B**) Comparison of the stalk region between the CA and TM domains of hCA IX (upper sequence) and mCA IX (lower sequence). The absence of three amino acids in the putative binding site of TACE/ADAM17 could impair the cleavage of the extracellular form of mCA IX. (**C**) Characterization of exosomes isolated from normoxic and hypoxic B16-F0 cells. Representative nanoparticle tracking analysis (NTA) of isolated exosomes diluted with PBS. The size distribution was analyzed by NTA using NanoSight NS500 (for each sample 5 × 60-s run). (**D**) mCA IX expression in exosomes isolated from normoxic and hypoxic B16-F0 cells determined by ELISA (Enzyme-linked Immunosorbent Assay). (**E**) Characterization of exosomes in parallel with cell lysates isolated from L-929 cells grown in normoxia and hypoxia for 48 h. Expression of mCA IX and CD63 (specific marker of exosomes) proteins was analyzed by Western blotting using specific antibodies. The anti-actin antibody was used as a loading control.

**Figure 4 ijms-21-00246-f004:**
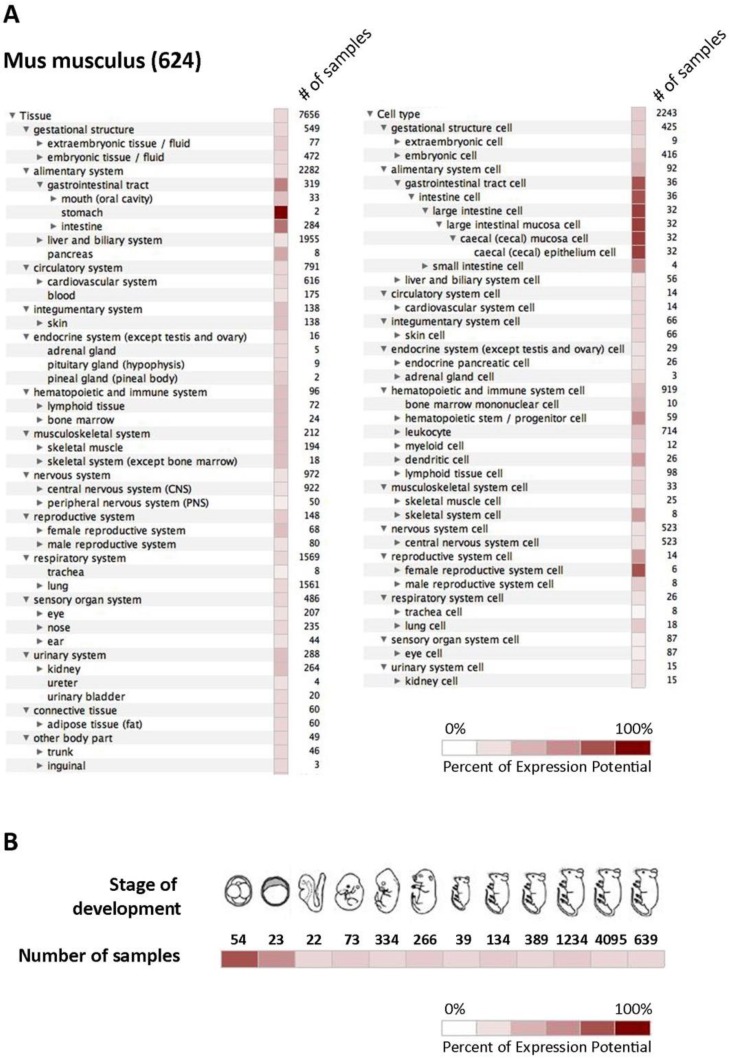
In silico analysis of mCA IX expression. Graphical output from Genevestigator analysis of *Car9* gene expression in mouse tissues (**A**) displayed by tissue (left side) and cell type (right side), as well as during twelve developmental stages (**B**). The expression potential is defined according to the results achieved by the analysis of a particular number of samples.

**Figure 5 ijms-21-00246-f005:**
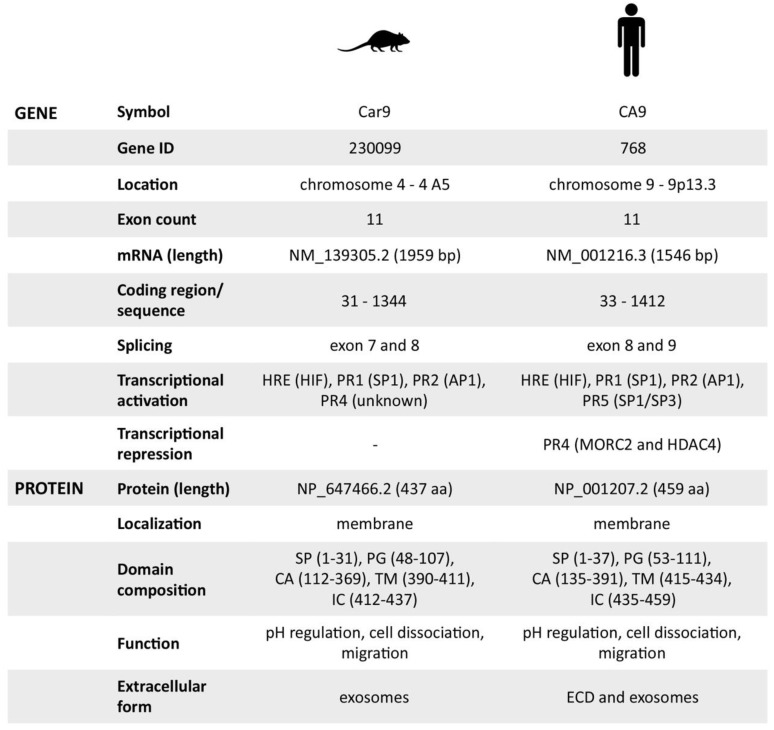
Overview of mouse and human carbonic anhydrase IX gene/protein characteristics. Summary of all previously mentioned characteristics, which are +/− similar for mouse and human.

**Table 1 ijms-21-00246-t001:** Sequences of the oligonucleotides used in this study. Sequences of the strands are written in the 5′ -> 3′ direction. Mutations against the wild-type sequence are indicated in bold.

Oligonucleotide	Sequence
MPR5Sac	TTTGAGCTCCTATGAACACACCTGCCCCTC
MPAXho	TTTCTCGAGAGCTGACTGGGGTGTCCCAGG
mHREMs	CCTATTTCCGATGC**TTT**TACAGCCCGTCCA
mHREMa	TGGACGGGCTGTA**AAA**GCATCGGAAATAGG
mSP1Ms	CAAGGCTTGCTCCT**AA**CCTACCCAGCTCCT
mSP1Ma	AGGAGCTGGGTAGG**TT**AGGAGCAAGCCTTG
mAP1Ms	AGGGCACTGTGAGT**TG**GCCTGCTCCCCTCA
mAP1Ma	TGAGGGGAGCAGGC**CA**ACTCACAGTGCCCT
mPR4s	GGGGAGGGGGCATAGGGCCAGACAGAACCTG
mPR4a	CAGGTTCTGTCTGGCCCTATGCCCCCTCCCC

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
