# Peer review of "Carbonic Anhydrase IX—Mouse versus Human"

_ijms, 2019, doi:10.3390/ijms21010246_

Round 1
Reviewer 1 Report
In this study, Takacova and colleagues report a characterization of mouse carbonic anhydrase IX expression. Using a combination of techniques and assays, they detail the transcriptional control of the Car9 promoter in response to normoxia and hypoxia, contrasting it with human CA9. They compare and contrast the mouse and human promoter regions in terms of transcription factor binding sites. They complement these data with functional studies on the consequences of the protein’s expression level, and establish the relative distribution of the mouse and human proteins between cellular, extracellular, and exosome fractions.
The authors make an important case that differential expression between mouse and human could impact interpretation of biological data, since carbonic anhydrase IX is a biomarker for indirect analysis of hypoxia in tumors. Thus, this work is significant and provides important information for the field.
The manuscript is well written and includes robust and overall well-presented data. There were a few instances where the manuscript could be further improved, which are noted below.
Major comments:
p.3, Fig. 1: In Fig. 1B, the experiments with the human CA9 promoter seem to have been done in only one condition. Could the authors clarify whether this was normoxia or hypoxia?
p.6, l.212: The migration data for L-929 cells is opposite to the predicted trend at short times – can the authors please provide an explanation for this?
p.7, l.242: The Western blot in Fig 3E does not show a CD63 band in the bottom right panel (for exosomes in normoxia), though it does show a band for mCA IX. Could the authors clarify why this is the case?
p.11: The discussion on p.11 could be condensed, in my view, since the data in the present manuscript do not address alternative splicing.
Minor comments:
p.2, l.76: “Relatively high identity and similarity…(75.8%)” – Could the authors provide the separate percentages for identity and similarity at this point, rather than later in the text?
p.10, l. 320: – “both proteins were simultaneously present in the PR4 sequence” – should this read “both protein binding sites”?
Author Response
We thank the reviewer for the favorable evaluation of our work.
p.3, Fig. 1: In Fig. 1B, the experiments with the human CA9 promoter seem to have been done in only one condition. Could the authors clarify whether this was normoxia or hypoxia?
We thank the reviewer for this notification. In fact, the human CA9 promoter was analyzed in both conditions, but the points illustrated in the graph represent a ratio of hypoxic to normoxic activity values to express fold-induction allowing for comparison to the mouse Car9 promoter.
For clarity, we included this information into the related Figure legend: “At the same time, the activity of human pGL3-CA9promoter was analyzed and is expressed as a fold induction (ratio of promoter activity in hypoxia compared to normoxia)”.
p.6, l.212: The migration data for L-929 cells is opposite to the predicted trend at short times – can the authors please provide an explanation for this?
This is a very good point, which can be explained by the fact that L-929 cells respond to hypoxia by decline in cell motility and migration (see Vogler et al, PLOS ONE, 2013, e69128). In our experiments, we wanted to overcome this effect and at the same time we needed to induce mCA IX expression to be able to evaluate its role in cell migration. Therefore, we pre-incubated L-929 cells in hypoxia for 48 h, and then placed them onto the migration platform in parallel with the normoxic controls. In the first phase of the migration experiment, L-929 cells needed to recover from the hypoxic pre-incubation and their migration was slower, but later on, these cells migrated faster than their fully normoxic counterparts, supporting the contribution of mCA IX.
To make the figure more intelligible, we changed the description of the graph, as well as the text in the legend to Fig. 2E.
p.7, l.242: The Western blot in Fig 3E does not show a CD63 band in the bottom right panel (for exosomes in normoxia), though it does show a band for mCA IX. Could the authors clarify why this is the case?
It was previously demonstrated by Gonzalez-King et al, Stem Cells 2017, 35:1747-1759, that CD63 is enriched in exosomes derived from hypoxic mesenchymal stem cells. There are also other studies indicating CD63 relationship to hypoxia in different cell types. We suppose that also in our case CD63 is enriched in hypoxic exosomes, and that it is actually not seen in the normoxic exosomes, because its level is under the detection limit by Western blotting.
p.11: The discussion on p.11 could be condensed, in my view, since the data in the present manuscript do not address alternative splicing.
Yes, we do agree with this comment and the information on the alternative splicing was condensed. See page 11, lines 342-345.
p.2, l.76: “Relatively high identity and similarity... (75.8%)” – Could the authors provide the separate percentages for identity and similarity at this point, rather than later in the text?
The percentages were identical and the information was included: “….identity and similarity between the human and mouse promoters (75.8% for both; Figure 1A), suggesting …..”.
p.10, l. 320: – “both proteins were simultaneously present in the PR4 sequence” – should this read “both protein binding sites”?
Thank you, we made the correction as proposed: “They confirmed that both protein binding sites were simultaneously present in the PR4 sequence, ….”.
Reviewer 2 Report
This is a good manuscript that compares CA-IX differences and similarities in mouse vs humans. In fact, mouse models are used for all the studies that is relevant to humans.This study shows that that in the case of CA-IX where mouse model is appropriate and where careful consideration is needed while using mouse models.This study will be very useful. I recommend publication of this article.
Author Response
We thank the reviewer for the favorable evaluation of our work.